# Ethnobotanical Knowledge on Herbs and Spices in Bulgarian Traditional Dry-Cured Meat Products

Teodora Ivanova [1,2,*], Mihail Chervenkov [1,2,3,*], Ekaterina Kozuharova [4] and Dessislava Dimitrova [1,2]

1   Department of Plant and Fungal Diversity and Resources, Institute of Biodiversity and Ecosystem Research, Bulgarian Academy of Sciences, 1113 Sofia, Bulgaria; dessidim3010@gmail.com
2   Slow Food in Bulgaria, 9 Pierre De Geytre St. bl. 3, 1113 Sofia, Bulgaria
3   Faculty of Veterinary Medicine, University of Forestry, 1797 Sofia, Bulgaria
4   Department of Pharmacognosy, Faculty of Pharmacy, Medical University-Sofia, 1000 Sofia, Bulgaria; ina_kozuharova@yahoo.co.uk
*   Correspondence: tai@bio.bas.bg (T.I.); vdmchervenkov@abv.bg (M.C.)

**Abstract:** Artisan food production, with its unique flavors, is a source of knowledge about sustainable use of natural resources. This is because it reflects the skills of local communities in utilizing these resources (e.g., wild and cultivated plants and autochthonous breeds) under specific environmental conditions for a long period of time. Therefore, the use of local ingredients and the reduction in food miles make traditional food a safer, healthier and more ecofriendly choice for consumers. In the present research, we examined the herbal ingredients in Bulgarian dry-cured meats and discuss their contribution to the flavor and durability of the products. A combination of field data, collected through semi-structured interviews in local communities, and an analysis of the available literature was used to reveal the typicity of 24 artisanal/homemade meat products available mostly in their place of origin. We compared the obtained data to 16 industrial products branded as "traditional", with 6 of them registered under the European Union quality schemes. The recorded ingredients of plant origin (dry and fresh) belonged to 16 taxa. Most diverse spice mixtures were used in products made of pork meat and in products originating from the southern, and hence warmer, regions of the country. The herbs and spices were combined freely except for *Alliums*, which were not mixed, and only one species was used per product. Most of the spices used in the artisanal dry meats were sourced from home gardens (some specifically cultivated for that purpose). Those collected from the wild, e.g., *Origanum vulgare* subp. *hirtum* (Link) Ietsw., were gathered sustainably in small quantities. The number of spices used in industrial dry meat products was limited to two–three, and was provided by cultivated sources, without exploiting natural populations. Manufacturing of all artisanal products was seasonal to avoid the cold winter weather, a measure which was necessary for the natural air-drying of the meat. The long-lasting effects of the abolishment of artisanal production under Communism, the adaptation of traditional products for industry, and the current challenges and perspectives surrounding artisanal production of meat products were discussed.

**Keywords:** ethnobotany; food plants; antispoilage; antimicrobial; traditional food; antioxidant

## 1. Introduction

Traditional food and the related knowledge reflect personal preferences, cultural upbringing and the local flora of the land, which contributes to the specific taste and aroma of these products [1]. This is especially valid for foods rich in proteins and fats, like those of meat and dairy origin, which often include diverse natural preservatives [2,3]. The fermentation caused by the spontaneous microflora of the raw ingredients and the environment contribute to the highly appreciated organoleptic characteristics and nutritional advantages of traditional dry-cured meat products [4–7]. Thus, fermented meat products are regarded as extremely valuable both culturally and commercially [8].

On the other hand, traditional methods of production are prone to contamination, originating both from the raw materials and the production premises, which are not always specifically arranged to limit or to reduce contamination with harmful microorganisms [9–11]. Various attempts have been made to improve the safety of traditional meat products; however, many of them do not meet the requirements of the customers to preserve the familiar taste and appearance of the products they have been accustomed to [12,13]. The use of diverse spices as natural antimicrobials in traditional meat products allows the reduction or exclusion of artificial preservatives, such as nitrates and nitrites, that are proven to be harmful for human health when used excessively [14]. These spices and herbs possess a wide spectrum of antimicrobial and antioxidant properties and in some cases their addition complements the effect of other added antimicrobials [15–18]. Additionally, herbal ingredients, exhibiting important biochemical properties, including antioxidant, anti-inflammatory, and anticarcinogenic properties, alleviate the burden of many noninfectious diseases, and hence they contribute to the overall health of the consumers [19–21]. Balanced and frequent consumption of bioactive compounds through food not only ensures improved availability but also boosts natural immunity and may drive beneficial alterations in gut communities, supporting digestion and bioabsorption [22–25].

So far, scientific studies on traditional meat products have not been definitive about local denominations, or they were commodified in a way that prevented comparisons and further exploration on their typicity and cultural value [26,27]. A lack of attention has also been given to the ingredients in regard to their origin and/or scientific taxonomy. While some local animal breeds have been the focus when studying heritage charcuterie [28,29], herbal ingredients have not been so frequently assessed from the (ethno)botanical perspective [30–32].

Due to the considerable economic transformations in the last 100 years, i.e., the abolishment of private property and the prevalence of industrial food production during the communist period, the authentic features of meat processing typical for Bulgarians have gradually faded away, and are therefore scarcely discussed in the scientific literature [33,34]. During the communist era (1945–1989), many recipes that originated from the Bulgarian gastronomic traditions were industrialized and modified by applying advances in food chemistry and technology, e.g., changing ingredients according to availability, adding preservatives and bulking agents and using artificial casings [35–37]. Studies on Bulgarian traditional meat products have focused predominantly on the industrial versions, dealing with safety issues, isolation of favorable microbial strains and a variety of approaches to improve quality and production technology [38–42]. Thus, local knowledge and traditions in the production of dry meat products were gradually assimilated and altered to fit an industrial production mode, focused mostly on quantities, but not necessarily following the food typicity and locality [33]. Hence, original, locally recognized meat products have remained solely within the personal/family domain of Bulgarians.

Nowadays, food naturalness, perceived as a proof for better quality, has gained importance for the consumers' choice and their increased interest in traditional food [43,44]. Positive reception of artisanal and homemade products has been registered at the growing number of farmers' markets, specialized events and within informal supply networks in Bulgaria.

Our study explores the data on traditional, artisanally manufactured, Bulgarian dry-fermented meat products, focusing on the ingredients of plant origin that reflect local flora and the related traditions, preserved and appreciated by the local communities. Additionally, we compare the ingredients of artisanal and industrial meat products, because some of the latter have more than 50 years of market history and have been registered as products with Protected Geographical Indication or as Traditional Specialty Guaranteed under EU quality schemes [45].

## 2. Materials and Methods

Bulgaria is located in the Balkan Peninsula, Southeastern Europe. The territory of the country (110,994 square kilometers) is in the transitional region between the Mediterranean

and temperate climatic zones, with slightly elevated average annual temperatures and lower precipitation rates in the last three decades [46]. Bulgarian vascular flora comprises 4064 species of spermatophytes affiliated with 921 genera and 159 families [47].

In the current study, we used a combination of methods: survey of archives, ethnographic sources, scientific literature, and analysis of field data obtained through semi-structured interviews in local communities. We explored the academic and popular literature published internationally and on a national level referring to dry-fermented meat products with known geographical origins. Data on ingredients and production methods were collected using: (i) scientific literature—Web of Science Scopus and AGRIS (FAO) databases and published books; (ii) recipes of products qualified as Protected Geographical Indication or traditional specialties included in EU eAmbrosia Food Register [48]; (iii) entries from the Ark of Taste e-catalogue of Slow Food that currently harbors about 160 meat products from the Mediterranean region and the Balkans [49]; (iv) traditional products for personal use and/or offered on specialized farmers' markets were additionally assessed during field work (2012–2020).

Semi-structured interviews were conducted with representatives of 72 households, engaged in home production of traditional meat products for personal consumption and/or for farmers' markets. A total of 25 female and 47 male participants aged between 50 and 86 years (50% over 65 years) were addressed by semi-structured interviews. Oral consent of every participant was received prior the interviews. The guidelines prescribed in the Code of Ethics of the International Society of Ethnobiology [50,51] were followed during the field study, and its compliance was confirmed by the Scientific Council of the Institute of Biodiversity and Ecosystem Research, Bulgarian Academy of Sciences, acting as independent institutional Ethics Board (Decision No. 6A/21/05/21).

We focused on herbal ingredients used in dry sausages and whole-muscle meats that rely on natural fermentation. Meat products consumed fresh and/or containing cooked/fried ingredients were not included. Reference herbarium specimens and/or image data of the presented herbal ingredients were collected for identification purposes and herbarium vouchers were deposited in the Herbarium of the Institute of Biodiversity and Ecosystem Research, Bulgarian Academy of Sciences (SOM). Identification of the obtained herbal ingredients was carried out at least to the species taxonomical level in accordance with Delipavlov et al. [52]. Data on spices and herbal ingredients used for curing of the different types of meat were presented via Venn diagrams and compared using the Jaccard (similarity) index (JI) for each pair of datasets. Calculation of the JI was performed using the following formula:

$$JI\,(X,\,Y) = |\,X \cap Y\,|\,/\,|\,X \cup Y\,|,$$

where X and Y signify every two datasets. JI ranges from 0 (no similarity) to 1 total equality [53].

## 3. Results

We report on herbal ingredients of 24 artisanal dry meat products documented during our field research in 8 provinces of Bulgaria, which were compared with the spice mixtures in 16 industrialized traditional products, of which 5 are registered as Traditional Specialty Guaranteed (TSG) and one as Protected Geographical Indication (PGI) under the European Union (EU) quality schemes (Figure 1, Supplementary Table S1).

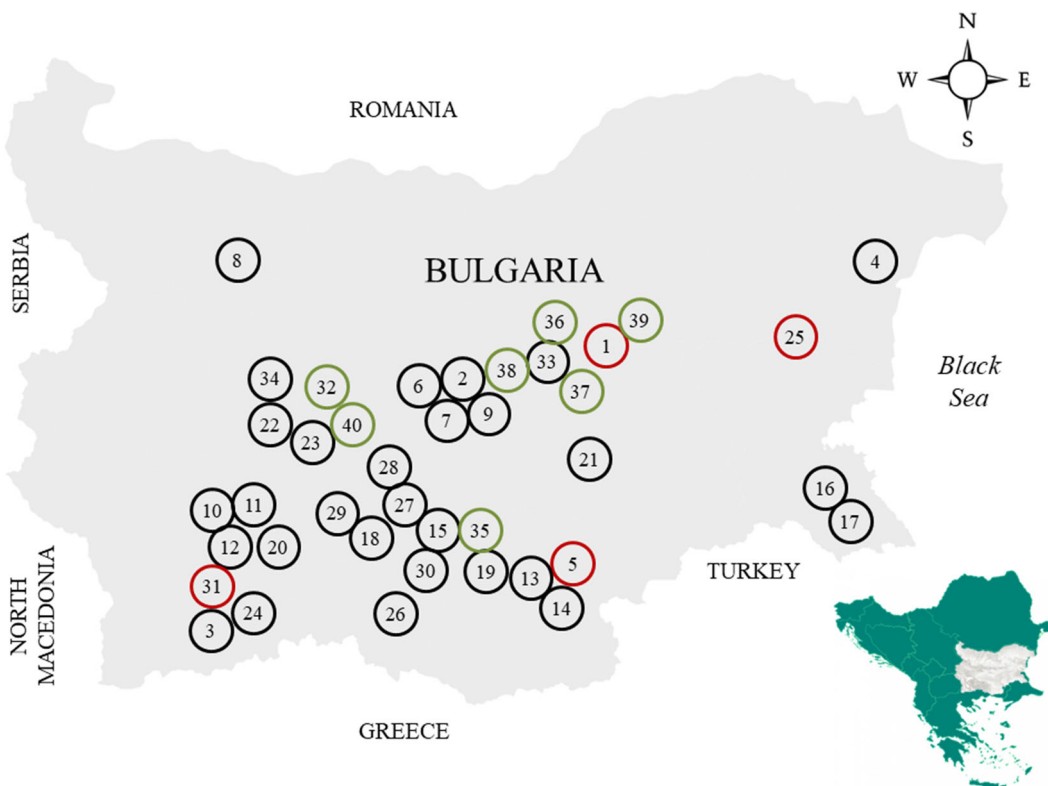

**Figure 1.** Provenance of the studied dry-cured meat products. * Bulgarian artisanal meat product industrially manufactured in the past (circled in red); ** industrialized version of Bulgarian traditional dry meat products registered as Traditional Specialty Guaranteed (TSG) or Protected Geographical Indication (PGI) (circled in green). 1—*Elenski* *; 2—*Etarska kaltsanitsa*; 3—*Ovcha pastarma*; 4—*Babichka*; 5—*Filevska puska* *; 6—*Gabrovsko file*; 7—*Gabrovska lukanka*; 8—*Kalbasa*; 9—*Konski sudzhuk*; 10—*Meurche*; 11—*Babichka*; 12—*Dedets*; 13—*Sarnachka puska*; 14—*Selska lukanka*; 15—*Starodzhelezarki starets*; 16—*Strandzhenska baba*; 17—*Strandzhensko dyado/parduhalche*; 18—*Tsalapishki babek*; 19—*Yabylkovska puska*; 20—*Tranenik*; 21—*Nozagorska lukanka*; 22—*Banska lukanka*; 23—*Banska karvavitsa*; 24—*Ovchi nogi*; 25—*Lukanka smyadovska* *; 26—*Kemik pastarma/Kakach*; 27—*Karlovska nadenitsa*; 28—*Karlovski babek*; 29—*Malokonarski babek*; 30—*Plovdivski starets*; 31—*Banski starets* *; 32—*Koprivshtenska lukanka* **; 33—*Velikotarnovska pushena lukanka*; 34—*Pirdopska lukanka*; 35—*Kayserovan vrat Trakiya* **; 36—*Role Trapezitsa* **; 37—*File Elena* **; 38—*Pastarma Govezhda* **; 39—*Gornooryahovski sudzhuk* **; 40—*Lukanka Panagyurska* **. Smaller image in the bottom right shows Bulgaria in the Balkan Peninsula.

Artisanal dry meat products occur mostly in mountain and semi-mountain areas where the climate favors the natural fermentation processes. The products were classified into three major groups—sausages (32), pastrami (7) (dry meat cuts and pastramis), ham (1)—and there was one that did not fit in any of the three categories. Over 86% of all products and half of the products found during the field study were sausages prepared of minced or chopped meat.

Respecting the traditions, manufacturing of all artisanal products was seasonal to avail from the cold winter weather, necessary for the natural air drying of the meat. In rural areas, the preparation of meat products was conducted mostly in December (also related to the Christmas traditions) when the slaughtering of backyard-raised animals for the family takes place. Whole-muscle pieces were typical for mountainous regions where cooler weather conditions allowed longer periods of drying indoors (mainly in attics) or in sheds in home gardens. Generally, meat was portioned according to the recipe, cured and hung to dry.

Herbal ingredients used in the Bulgarian traditional meat products documented during the field studies, were mostly locally sourced—collected from the wild, grown in the home garden or bought at farmers' markets (Table 1). The latter was the case mainly in urban areas. Most of the documented food plants (14) were applied as dry spices and only *Allium* species were added as chopped fresh bulbs or pseudostems and leaves (Figure 2).

**Table 1.** Culinary plants used for the preparation of Bulgarian dry-cured meat products.

| Family | Species | English Name | Parts Used | Source |
|---|---|---|---|---|
| Amaryllidaceae | *Allium ampeloprasum* L. | Leek | leaves and pseudostem | home gardening/farmers' markets |
| Amaryllidaceae | *A. sativa* L. | Garlic | Bulb | home gardening/farmers' markets |
| Amaryllidaceae | *A. cepa* L. | Onion | Bulb | home gardening/farmers' markets |
| Apiaceae | *Cuminum cyminum* L. | Cumin | Seeds | imported |
| Apiaceae | *Coriandrum sativum* L. | Coriander | Seeds | home gardening/wild |
| Apiaceae | *Foeniculum vulgare* L. | Fennel | Seeds | home gardening/wild |
| Piperaceae | *Piper nigrum* L. | Black pepper | fruit (drupe) | imported |
| Solanaceae | *Capsicum annuum* L. | Chili pepper/Paprika (sweet) | powdered fruits and seeds or flakes | home gardening/farmers' markets |
| Solanaceae | *Capsicum annuum* L. | Chili pepper (hot) | powder fruits and seeds or flakes | home gardening/farmers' markets |
| Lamiaceae | *Satureja hortensis* L. | Savory | Herbage | home gardening/farmers' markets |
| Lamiaceae | *Origanum vulgare* L. ssp. *hirtum* (Link) Ietsw. | Oregano | Herbage | wild, home gardening |
| Brassicaceae | *Capsella bursa-pastoris* L. | Shepard's purse | Herbage | wild |
| Fabaceae | *Trigonella foenum-graecum* L. | Fenugreek | Seeds | home gardening/farmers' markets |
| Myrtaceae | *Pimenta dioica* (L.) Merr. | Allspice | Fruit | imported |

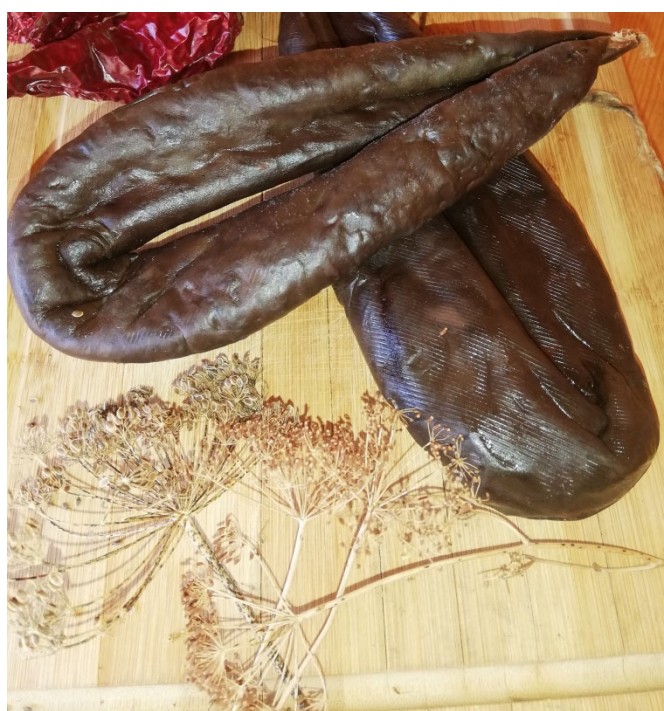

**Figure 2.** *Tranenik* sausage (total length of ca. 55–60 cm) made of raw pork liver with leek (*Allium ampeloprasum* L. and local "pork spice" (*Coriandrum sativum* L.) from Gorno Draglishte village (Blagoevgrad province).

Additionally, when *Allium* is used for spicing, only one species is used, and it has never been mixed with other representatives of Amaryllidaceae. For example, when leak is used, they will not add onion or garlic in the same product. Leak (*Allium ampeloprasum* L.) was the most frequently used species from this genus (found in six products). It was never added to beef and mutton/goat meat products and was found to be the only common herb in pork and horse meat products (JI = 0.23, Figure 3).

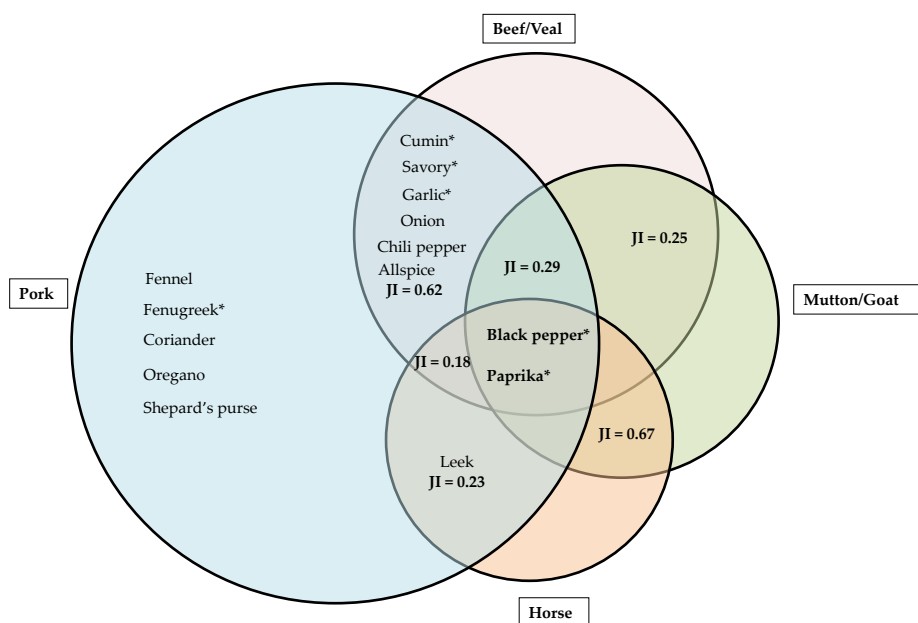

**Figure 3.** Comparison of the plant-based ingredients used for curing of different types of meat in artisanal Bulgarian traditional air-dried meat products. * Ingredients used in industrial versions. JI—Jaccard similarity index.

Chopped fresh leek was added in pork sausages mainly in South Bulgaria. Leek was also found in two sausages from the area of Gabrovo—*Etarska kaltsanitsa* and *Konski sudzhuk*, the latter being the only product containing horse meat. Onion (*Allium cepa* L.) was rarely found—only in the *Novozagorska lukanka* and in the *Kalbasa* of the Banat Bulgarians, a Catholic minority from North Bulgaria.

The pork meat products were characterized by more diverse mixtures of dry spices, the diversity being even higher in artisanal/homemade pork sausages (Figure 4). While some of the spices, such as savory (*Satureja hortensis* L.) and sweet paprika powder (*Capsicum annuum* L.), were popular, and the most often used herbal ingredients of local origin all over the country, others were restricted to specific areas, e.g., shepherd's purse (*Capsella bursa-pastoris* L.) was used in *Strandzhensko dyado* and *parduhalche* from Strandzha Mt.; oregano (*Origanum vulgare* L. ssp. *hirtum*) was used in *Filevska puska* from Haskovo province (Upper Thracian Valley); fenugreek (*Trigonella foenum-graecum* L.) was used in sausages from Gabrovo province (Central Balkan Mts.); and coriander (*Coriandrum sativum* L.) was used in *navpavok* sausages from Razlog valley (Rila Mt.).

Black pepper and cumin were the most frequently used dry spices, in 26 and 18 products, respectively. Both spices were bought as whole seeds or ground powder from the shops. In artisanal/homemade products, the quantities of the two spices (and this of the salt) were more frequently measured in the traditional manner, i.e., in the number of full match boxes, rather than using the precise net weight. The quantities of the remaining (locally sourced) spices were measured visually—"by eye"—but not by taste, which results in the creation of diverse individual- or family-specific tastes that are publicly discussed and compared within the community. Visual apprehension of the spice quantities and ratios was strictly individual. Presence or absence of a certain spice and spice ratios used in the spice mixtures was discussed, usually during celebratory diners, as a part of the artisanship. The makers of the most praised sausages become famous in their communities as "masters".

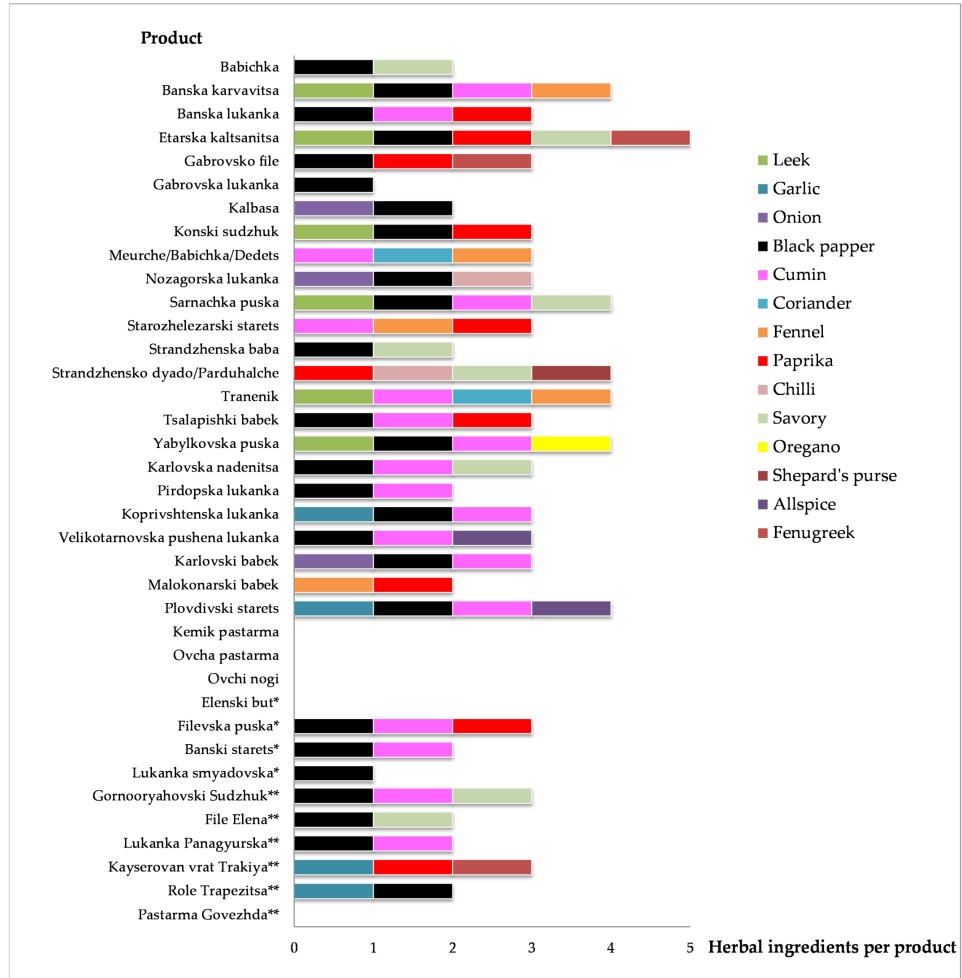

**Figure 4.** Herbal ingredients in Bulgarian traditional dry-cured meat products. * Bulgarian artisanal meat product industrially manufactured in the past; ** industrialized versions of Bulgarian traditional dry meat products registered as Traditional Specialty Guaranteed (TSG) or Protected Geographical Indication (PGI).

Black pepper (*Piper nigrum* L.) and paprika powder or flakes were used in products made of all types of meat. Five spices, namely fennel (*Foeniculum vulgare* L.), fenugreek, coriander, oregano and shepherd's purse were specifically used in pork artisanal/homemade products, but were absent in products made of other types of meat. A notable example here was *Navpavok*, a collective name for the three different sausages from Razlog valley (*Meurche*, *Babichka*, *Dedets*). Coriander was considered of central importance for the *Navpavok*, it is called *gudecha merudia* (pork spice in the local dialect), and is cultivated in the rural home gardens solely for the preparation of pork delicacies.

*Foeniculum vulgare* L. was similarly important for the pork sausages prepared in villages along Maritsa River in Central Southern Bulgaria, where fennel is cultivated in home gardens and is used for the preparation not only of dry fermented sausages but also in vegetable preserves and flavored salts, as well as being added to different dishes. These villages are famous for pepper growing, and locals rely also on homemade sweet paprika powder and chili flakes for the specific flavor of their sausages. Some of these sausages (*Tsalapishki babek* and *Malokonarski babek*) were also additionally smoked during the drying phase so to improve the flavor and their storability.

Beef/veal sausages were cured with very similar spice mixtures used in pork products (JI = 0.62), while mutton/goat meat products might not contain any herbal ingredients. According to our respondents, the homemade products containing fewer or no herbal ingredients are among those that are most threatened by abandonment, as modern refriger-

ation tends to replace lengthy air-drying preservation in ambient conditions. Additionally, milder winters in recent decades have compromised the production of products both with and without natural herbal ingredients, since safe aging of the meat cannot be ensured.

Compared with the artisanal/homemade products included in our study, industrial versions holding the Traditional Specialty Guaranteed (TSG) label contained only pork, beef or combination of both, and offered a limited range of flavors—five dry spices and garlic (*Allium sativum* L.) as a seasoning. While in artisanal/homemade products, 62.5% of the products contained three or more herbal ingredients, among the industrialized meat products with the TSG label, only three had more than two added spices. Similar to the artisanal/homemade products, black pepper and cumin were the dominating spices in the industrialized traditional products. Garlic was found in the recipes of two of the seven TSG industrial products (*Lukanka Panagyurska* and *Kayserovan vrat Trakiya*). *Gornooryahovski sudzhuk*—a beef sausage that is the only Bulgarian meat product registered with the Protected Geographical Indication (PGI)—was flavored with black pepper and cumin, but also with savory (*Satureja hortensis* L.), which otherwise was found only in artisanal/homemade pork products. Similarly, in the industrialized products, cumin was found both in pork and beef products, while in the artisanal/homemade products, it was noted only in sausages made of pork meat. According to our participants, imported cumin seeds have replaced those previously collected from wild caraway (*Carum carvi* L.) for the traditional dry meats. However, imported cumin, offered commonly in almost every store, was preferred out of convenience.

## 4. Discussion

There is an increasing tendency toward the consumption of red meat in Bulgaria, which is currently maintained by the rising import of fresh and frozen meat and processed meat products [54]. Our latest research has also shown that Bulgarians have one of the highest preferences for red meat over poultry and other white meats in Southern Europe [55,56]. This was well reflected by the number of traditional meat products made of red meats, presented in the current study. The number of herbal ingredients and their combinations were the most diverse in traditional meat products made of pork—the preferred meat type of most of the Bulgarian participants in this study. Family recipes, handed down through generations, and the availability of ingredients, especially in the rural areas, have allowed the preservation of the knowledge for the preparation of dry-fermented meat products all over the country.

Most of the currently assessed herbal ingredients are popular in the cuisines of other Mediterranean and Balkan countries as well, and even imported ones such as black pepper and cumin are regarded as "traditional" and are commonly used in the preparation of traditional meat products around Europe [14,57–60]. Herbal ingredients in Bulgarian traditional meat products rarely included oregano, rosemary or thyme, which are commonly used for meat flavoring around Europe [61]. Vignolo et al. suggested that the combination of garlic, cumin and black pepper was responsible for the typical flavor of "Bulgarian *loukanka*", a generic description of one of the industrialized types of sausages not related to any specific region in Bulgaria [62]. Since the industrial versions were and still are more easily available to foreign consumers, the aforementioned pepper–cumin–garlic flavor formula, typical for "Bulgarian *loukanka*", has probably appropriated a "signature role" in the Bulgarian tradition in dry-cured meat production. However, our field observations failed to spot artisanal/homemade products with such a combination of spices. On the contrary, only 13 of the studied artisanal/homemade products contained any *Allium* in the recipe. Overall, the artisanal/homemade products more often contained three or more herbal ingredients, which could be related to the quest for antimicrobial and antiradical activity contained by the essential oils and other phytochemicals [3,63–66]. Studied industrial versions of Bulgarian traditional meat products contained simpler combinations of two–three herbs and/or spices, used both in pork and veal meat products. While such an approach ensures more convenient ingredient supply and safety management, it hardly allows the creation of

unique memorable flavors that represent local tradition and terroir. For instance, historical studies on the artisanal manufacturing of *Gornooryahovski sudzhuk*, one of the popular and frequently exported Bulgarian food products, revealed a variety of recipes recorded before 1944, with different numbers and combinations of spices, including nutmeg, allspice, cinnamon, etc. [67]. However, nowadays, the recipe for *Gornooryahovski sudzhuk* (prepared only from veal), registered as PGI, has only three spices—cumin, savory and black pepper.

Some traditional products made for personal use were found to contain large amounts of salt, nitrates and nitrites, and thus more efforts are needed to raise the awareness of the producers for using traditional recipes on topics related to healthy lifestyle [68]. Additionally, the promotion of the use and combination of different spices and preservation techniques can ensure the production of safer and healthier products. Usage of spice mixtures together with *Alliums* implied that Bulgarian traditional production of dry-cured sausages relied on the herbal ingredients, not only for their organoleptic features but also for their antibacterial, antifungal and insecticide properties [69,70]. This was especially important in the warmer southern parts of the country, where we found five of the seven meat products with four herbal ingredients that are known for their antimicrobial and antifungal properties [15,18,71]. Leek was preferred for the preparation of traditional sausages, a tradition also popular in northern parts of Greece. Leek contributes to the preservation of the fresher red color of the meat after drying due to the high nitrate content, especially in the pseudostem [72]. Onion was less popular, present in *Novozagorska lukanka* and *Kalbasa*. The latter shared some similarities with the Polish smoked *Kielbasa* and Slovenian *Kranjska klobasa*, which typically contain garlic instead of onion [73,74]. Natural phenolic compounds found in many *Alliums* and Lamiaceae species possess considerable antibacterial, antifungal and antioxidative activities that have made them very popular in meat processing [61].

The observed usage of three or more spices that are mixed with the sausage meat is not so typical for preserved meats produced in Europe, Africa and the Middle East [8,14,62,75–77]. *Capsella bursa-pastoris* could be regarded as the most unusual aromatic herb, used locally in Strandzha Mt., Southeast Bulgaria, in the preparation of pork sausages. Shepard's purse is known as a leafy vegetable in other cultures and it is recognized for its antimicrobial and antioxidant properties [78–80]. However, this is its first mention in the preservation of meat products, which requires additional exploration. On the other hand, the most common spices used in Bulgarian traditional meat products were black pepper and paprika powder. They are important not only as taste and flavor enhancers, but also because of their health benefits. Black pepper is a universally known spice and medicinal plant with antimicrobial, antifungal and anti-inflammatory effects, due to the rich variety of piperidines, phenolics, flavonoids, proanthocyanidins, etc. [81,82]; meanwhile, an oil-soluble fraction of paprika powder contains predominately capsaicin, capsanthin and capsorubin, which, as a complex, contribute to the safety of the traditional products, as strong antioxidant and antimicrobial agents, while bringing its natural flavoring and coloration [83]. Savory and paprika powder, the most often used herbal ingredients of local origin in Bulgarian dry-cured products, are traditionally used in several European meat products with protected geographical origin (e.g., Iberian Chorizo, Slavonian Kulen and Morcilla de Burgos and others) [60,84,85]. *Satureja* species are among the popularly used herbs for seasoning of fresh and cooked meat products in some parts of Southern Europe and Western and Middle Asia due to their pronounced antimicrobial activity, attributable to the high content of carvacrol and thymol [86,87]. However, *S. montana* L., that is used in the Balkans and in Spain, is currently added to the meat products as an essential oil and/or extract [88]. Both were reported to express strong antioxidant and antibacterial activity against *Salmonella*, *Listeria* and *Staphylococcus* due to the presence of carvacrol and *p*-cymene [88,89]. In Bulgarian traditional meat products, dry *S. hortensis* herbage was found in recipes with both pork and/or beef/veal meat. Main constituents of *S. hortensis* essential oil (carvacrol, thymol, $\gamma$-terpinene and *p*-cymene) were reported to vary significantly depending on a number of factors, such as climate, genetic factors, harvesting times, etc., with carvacrol being

the most important element for the antimicrobial activity of the essential oils [87,90,91]. Overall, the aromatic composition and major constituents of *Satureja* oils show significant intra- and interspecific variation [92]. Therefore, the foreign savory spices or similar of unknown origin could easily compromise the typical features of the final products, which underlines the importance of locally sourced ingredients for traditional products, especially when they are produced in greater volumes for the market [93]. Another carvacrol-rich herbal ingredient in the Bulgarian traditional sausages, far less frequently used than savory, is *Origanum vulgare* ssp. *hirtum* (Link) Ietsw. Carvacrol and thymol are one of the most important compounds in oregano essential oils that contributes to its strong antibacterial and antioxidative effect and has made it popular in the meat processing industry in the recent years [94,95]. Inter- and intra-populational diversity of the essential oil compounds of the Bulgarian populations of *O. vulgare* ssp. *hirtum* was found to be high, with most of the plants belonging to the carvacrol chemotype [96,97]. The collection of *O. vulgare* ssp. *hirtum* for the market is currently banned in Bulgaria due to its limited distribution in the country and the overexploitation of the wild populations [98]. Small-scale traditional food production in Bulgaria fostered the domestication of the threatened *O. vulgare* ssp. *hirtum* in home gardens, which contributes to the alleviation of the pressure on the natural populations [99].

In the artisanal/homemade pork meat products, the number of herbal ingredients included varied from one to five, and only one product had no herbal ingredients in the curing mixture (*Elenski but* ham). Products made of beef/veal, buffalo, horse or mutton meat, and especially the whole-muscle cuts, had fewer herbal ingredients. Topical application of pungent dry herbs and spices is more usual for the dry whole-muscle cuts that are first salted and then covered with specific mixes [8,100,101]. Traditional Bulgarian products from whole-muscle meat cuts are prepared in a similar way to the pastirma/bastirma method known also from the Middle Eastern and Mediterranean regions [8,77]. However, the currently assessed products made of mutton/lamb and beef, favored in Muslim communities, were predominantly cured with plain sea salt without application of any spices or herbs. The latter implies that they were probably destined for further culinary preparations. According to ethnographic data from first half of the twentieth century in the Central Balkan Mts., *pastarma* products, made of mutton, goat and buffalo meat, prepared by dry- and wet-curing in brine, were also used for the preparation of various dishes [102].

The same preservation method is used for the studied whole-muscle pork meat products; however, those were almost always flavored with spice mixtures. In this sense, the addition of herbs and spices categorizes these traditional meat products as products of higher value, as delicacies or mezze (starters), served and consumed in a celebratory manner [103,104]. This could be also related to the traditional practices of presenting sausages and other valuable traditional foods as part of the customary expressions of gratitude and appraisal during different occasions, which was practiced by some of our respondents. In this sense, Bulgarian artisanal and homemade dry-cured meat products cannot be considered as processed products that utilize low-quality cuts and other nonanimal ingredients, as they are in other European countries [7], but as added-value products.

## 5. Conclusions

Understanding local food knowledge and gastronomic traditions gives insights to sustainable agricultural practices and food processing. Given the very dynamic changes in health, socioeconomic situations and climate, occurring both in Bulgaria and on a global scale, upholding this knowledge and the related traditions is a critical issue both for remote communities and in fast-depopulating countries alike to Bulgaria. Discerning characteristic features of locally preserved food products that reflect local wild floras and agrobiodiversity is an important tool to identify current and potential threats, to push suitable incentives for their preservation, and to raise awareness of important themes, such as climate change and nature protection, through comprehensible messages for the general public. In countries that have undergone considerable transformation in food production

and economy, rigorous exploration of local traditional food and related knowledge for the collection of wild herbs and spices—as well as collection programs for locally cultivated food plants—is needed to preserve the reducing number of guardians of these resources.

**Supplementary Materials:** The following supporting information can be downloaded at: https://www.mdpi.com/article/10.3390/d14060416/s1, Supplementary Table S1: Provenance and meat type of Bulgarian traditional dry-cured meat products.

**Author Contributions:** Conceptualization, T.I. and M.C.; methodology, T.I.; investigation, D.D., T.I., E.K. and M.C.; writing—original draft preparation, review and editing, T.I., M.C., E.K. and D.D. All authors have read and agreed to the published version of the manuscript.

**Funding:** This study has been carried out in the framework of the National Science Program "Environmental Protection and Reduction of Risks of Adverse Events and Natural Disasters", approved by the Resolution of the Council of Ministers No. 577/17.08.2018 and supported by the Ministry of Education and Science (MES) of Bulgaria (Agreement No. D01-230/6 Decemeber 2018).

**Institutional Review Board Statement:** The study was conducted in accordance ethical principles prescribed in the Code of Ethics of the International Society of Ethnobiology and its compliance was confirmed by the Scientific Council of the Institute of Biodiversity and Ecosystem Research, Bulgarian Academy of Sciences, acting as independent institutional Ethics Board (Decision No. 6A/21/05/21).

**Informed Consent Statement:** Informed consent was obtained from all participants involved in the study.

**Data Availability Statement:** Data can be found within the paper.

**Acknowledgments:** Authors are grateful to all participants in the field research, the rightful owners of this traditional knowledge, for the shared experience and provided samples. Generous help and collaboration of the local supporters of Slow Food movement in Bulgaria are also appreciated.

**Conflicts of Interest:** The authors declare no conflict of interest.

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
