# Peer review of "Ethnobotanical Knowledge on Herbs and Spices in Bulgarian Traditional Dry-Cured Meat Products"

_diversity, doi:10.3390/d14060416_

Round 1

Reviewer 1 Report

Manuscript ID: diversity-1703462

Overall Recommendation:

Minor Revision suggested

General Remarks

The objective of the work entitled “Ethnobotanical knowledge on spices in Bulgarian traditional dry cured meat products” was to investigate and record the diversity of herbs and spices traditionally used for the making of cured meat products in Bulgaria. For this reason, data were collected from scientific and popular literature, as well as sufficient data were provided by an eight-year long field work. Moreover, a straightforward statistical analysis was performed to portray the most important findings of the study.

The manuscript is concise and well-organized. The English language and writing style are adequate for publishing. The introduction is well-written, providing background information for the research and also a clear statement of the purpose of the work. The results and discussion parts are likewise presented in a reader-friendly way and are easy to follow. Additionally, the authors cite sufficient literature data for the reader to better understand and clarify the key points concerning their research findings.

The research is suitable for publication in Diversity. I think that the manuscript should be accepted post the recommended minor revision, and that the findings will be useful for the scientific community.

Specific Comments to the Authors
  • Title: I would suggest transforming the title to:

Ethnobotanical knowledge on herbs and spices in Bulgarian traditional dry cured meat products”

  • Lines 52-58: Herbs and spices serve as antimicrobials and/or antioxidants during the making of traditional meat products but also in the final foodstuffs thus contributing to their preservation. Moreover, phytochemicals found in herbs and spices exhibit important biochemical properties, including antioxidant, anti-inflammatory etc. for the consumer. The authors do make a short report on the above-mentioned properties, but I suggest that the dual importance of using spices to be a bit more highlighted and clear in the text.

  • Lines 106-107: Please add the appropriate citations and references for eAmbrosia and Arc Taste, including website name, url and last access date.

  • Line 130: It would be to better to change “...from 0 – no similarity to 1 – total” to “from 0 (no similarity) to 1 (total)”.

  • Figure 1: Please change TURKEY/TURKIYE to TURKEY, or else add the names for the other countries in order to keep uniformity in the map.

  • Table 1: Please change “Food plants...” to “Edible plants...” or “Culinary plants...”.

  • Line 290: Please form “et al.” in italics.

  • Line 306: The importance of preserving the culinary tradition of Bulgaria even in mass production scale consists a well-explained and underlined aspect of the study. However in my opinion, the term “disrespect” assigned to industrial practices performed by meat technologists is rather rigid and mordant. I suggest changing the specific part of the text, as to depict only the pure fact, and maybe to reflect a real purpose of this study which could be to set the basis and a form of guidance addressed to the meat industry in order to align the making of their products with the traditional Bulgarian recipes.

  • Supplementary table: Please insert a reference for eAmbrosia.

Author Response

Dear Reviewer,

Thank you for the positive recommendations and for the careful consideration to our work. We are grateful for the detailed reviews that will improve the quality and readability of the current manuscript. We tried our best to correct the pointed-out insufficiencies and to integrate your suggestions. Please, find bellow the responses to the specific comments.

  • Title: I would suggest transforming the title to:

“Ethnobotanical knowledge on herbs and spices in Bulgarian traditional dry cured meat products”

Thank you for the suggestion. We changed the title accordingly.

  • Lines 52-58: Herbs and spices serve as antimicrobials and/or antioxidants during the making of traditional meat products but also in the final foodstuffs thus contributing to their preservation. Moreover, phytochemicals found in herbs and spices exhibit important biochemical properties, including antioxidant, anti-inflammatory etc. for the consumer. The authors do make a short report on the above-mentioned properties, but I suggest that the dual importance of using spices to be a bit more highlighted and clear in the text.

We appreciated this suggestion very much and extended this paragraph with the following:

These spices and herbs possess a wide spectrum of antimicrobial and antioxidant properties and in some cases their addition complements the effect of other added antimicrobials [15–18]. Additionally, herbal ingredients, exhibiting important biochemical properties, including antioxidant, anti-inflammatory, anticarcinogenic, etc., alleviate the burden of many non-infectious diseases; hence contribute to the overall health of the consumers. [19–21]. Balanced and frequent consumption of bioactive compounds through food not only ensures improved availability but also boosts natural immunity and may drive beneficial alterations in gut communities supporting digestion and bioabsorption [22–25].

  • Lines 106-107: Please add the appropriate citations and references for eAmbrosia and Arc Taste, including website name, url and last access date.

Reference list is updated.

  • Line 130: It would be to better to change “...from 0 – no similarity to 1 – total” to “from 0 (no similarity) to 1 (total)”.

Correction is made in the text.

  • Figure 1: Please change TURKEY/TURKIYE to TURKEY, or else add the names for the other countries in order to keep uniformity in the map.

Thank you for this remark! Turkey has announced an official state name change with the UN, however, this change is not relevant for the study and we will keep only the current one.

  • Table 1: Please change “Food plants...” to “Edible plants...” or “Culinary plants...”.

Changed to Culinary plants.

  • Line 290: Please form “et al.” in italics.

Corrected.

  • Line 306: The importance of preserving the culinary tradition of Bulgaria even in mass production scale consists a well-explained and underlined aspect of the study. However in my opinion, the term “disrespect” assigned to industrial practices performed by meat technologists is rather rigid and mordant. I suggest changing the specific part of the text, as to depict only the pure fact, and maybe to reflect a real purpose of this study which could be to set the basis and a form of guidance addressed to the meat industry in order to align the making of their products with the traditional Bulgarian recipes.

Thank you for this suggestion. We propose the following change of this section:

Studied industrial versions of Bulgarian traditional meat products containeda simpler combinations of two-three herbs and/or spices, used both in pork and veal meat products. While such approach ensures more convenient ingredient supply and safety management, it hardly provides for the creation of unique memorable flavours that represent local tradition and terroir. For instance, historical studies on the artisanal manufacturing of Gornooryahovski sudzhuk, one of the popular and frequently exported Bulgarian food products, revealed a variety of recipes recorded before 1944 with different number and combinations of spices including nutmeg, allspice, cinnamon, etc. [66]. However, nowadays the recipe for Gornooryahovski sudzhuk (prepared only from veal) and registered as PGI has only three spices – cumin, savory and black pepper..

We are grateful for the highlighted omissions. The manuscript was thoroughly re-checked for technical and grammar errors.

Sincerely,

Ivanova and co-authors.

Reviewer 2 Report

The manuscript is very interesting, addressing a novel piece of information. I have no hesitation to recommend it for publication in this reputed journal. The philosophy of this work is very good, and applicable in basic and applied plant sciences. The structure and content of the paper are understandable. There are some minor grammatical errors in the text, which can be removed with one careful reading by someone having expertise in the English language. To improve readership and the impact of this work, I would like to suggest one schematic diagram in the discussion of results.

Arguments need clearer and tighter presentation and more developed discussion.

The authors studied the plant ingredients in Bulgarian dried meat and discuss their contribution to the taste and durability of the products. The authors present interesting research. Spices are currently perceived not only in terms of taste but also the content of bio-ingredients as products of the secondary metabolism of plants. Please indicate which kind of spices, and metabolic components, supply the meat with improved quality in terms of food quality.

What is the authors' opinion on the use of medicinal herbs in combination with meat? What is the opinion of authors to use antiviral, antibacterial species?

I recommend that minor revisions are done to the article before publication. The results can promote future research.

Author Response

Dear Reviewer,

Thank you for the positive recommendations and for the careful consideration to our work. We are grateful for the detailed reviews that will improve the quality and readability of the current manuscript. We tried our best to correct the pointed-out insufficiencies and to integrate your suggestions. Please, find bellow the responses to the specific comments.

To improve readership and the impact of this work, I would like to suggest one schematic diagram in the discussion of results.

Thank you for the suggestion. We will consider suitable graphical abstract for this purpose.

Please indicate which kind of spices, and metabolic components, supply the meat with improved quality in terms of food quality. What is the authors' opinion on the use of medicinal herbs in combination with meat? What is the opinion of authors to use antiviral, antibacterial species?

We appreciated your recommendation and extended the discussion with the following text:

The most common spices used in Bulgarian traditional meat products were black pepper and paprika powder. They are important not only as taste and flavor enchancers but also because of their health benefits. Black pepper is universally known spice and medicinal plant with antimicrobial, antifungal, anti-inflammatory effects due to the rich variety of piperidines, phenolics, flavonoids, proanthocyanidins, etc. [80,81], while oil-soluble fraction of paprika powder contains predominately capsaicin, capsanthin and capsorubin that as a complex contribute for the safety of the traditional products, being strong antioxidant and antimicrobial agent, and also for the natural flavouring and coloration [82].

We are grateful for the highlighted omissions. The manuscript was thoroughly re-checked for technical and grammar errors.

Sincerely,

Ivanova and co-authors.

Reviewer 3 Report

In the current manuscript, the authors documented herbal ingredients in Bulgarian dry cured meats and discuss their  contribution to the flavor and durability of the products. This certainly is an interesting research that document and focuses on the perservation on an important indigenous knowledge in the study area. The methods are adequately described and appropriate to address the research problem. Overall, the presentation meets standard expected. However, a few concerns need to be addressed by the authors.

COMMENTS

Ensure you consult the guideline paper below to ensure your manuscript have met the acceptable standard.

Weckerle, C.S., de Boer, H.J., Puri, R.K., van Andel, T., Bussmann, R.W., Leonti, M. 2018. Recommended standards for conducting and reporting ethnopharmacological field studies. Journal of Ethnopharmacology 210:125-132.

One major issue is the absence of voucher specimens for the documented plants, have these plants been positively identified? The importance of this cannot be overephasised as artciulated in the ref suggested above.

RESULT: Currently verbose and need to be presented in a concise manner. Avoid repeating obvious description and rather give stand-out for readers

DISCUSSION: Avoid unnecessary fragmentations, a number of the paragraphs are too short and need to be properly integrated to enhance the flow of the discussion.

Finally, See ATTACHED PDF for minor comments in track changes.

Author Response

Dear Reviewer,

Thank you for the positive recommendations and for the careful consideration to our work. We are grateful for the detailed reviews that will improve the quality and readability of the current manuscript. We tried our best to correct the pointed-out insufficiencies and to integrate your suggestions. Please, find bellow the responses to the specific comments.

Ensure you consult the guideline paper below to ensure your manuscript have met the acceptable standard.

Weckerle, C.S., de Boer, H.J., Puri, R.K., van Andel, T., Bussmann, R.W., Leonti, M. 2018. Recommended standards for conducting and reporting ethnopharmacological field studies. Journal of Ethnopharmacology 210:125-132.

One major issue is the absence of voucher specimens for the documented plants, have these plants been positively identified? The importance of this cannot be overephasised as artciulated in the ref suggested above.

Thank you for this remark! Information on the identification and documentation procedures of plant specimens was unintentionally omitted and is now restored in the Material and Methods section:

Reference herbarium specimens and/or image data of the presented herbal ingredients were collected for identification purposes and herbarium vouchers were deposited in the Herbarium of the Institute of Biodiversity and Ecosystem Research, Bulgarian Academy of Sciences (SOM). Identification of the obtained herbal ingredients was carried out at least to species taxonomical level in accordance with Delipavlov et al. [51].

RESULT: Currently verbose and need to be presented in a concise manner. Avoid repeating obvious description and rather give stand-out for readers. DISCUSSION: Avoid unnecessary fragmentations, a number of the paragraphs are too short and need to be properly integrated to enhance the flow of the discussion.

We appreciate the proposed optimizations. We propose a slight rearrangement of the paragraphs so to integrate some of the shorter ones.

Finally, See ATTACHED PDF for minor comments in track changes.

We are grateful for the highlighted omissions. The manuscript was thoroughly re-checked for technical and grammar errors.

Sincerely,

Ivanova and co-authors.